# Bioaccumulation and Mobility of Heavy Metals in the Soil-Plant System and Health Risk Assessment of Vegetables Irrigated by Wastewater

Muhammad Tansar Abbas [1], Mohammad Ahmad Wadaan [2], Hidayat Ullah [1,*], Muhammad Farooq [3], Fozia Fozia [4,*], Ijaz Ahmad [5], Muhammad Farooq Khan [2], Almohannad Baabbad [2] and Zia Ullah [6]

1    Institute of Chemical Sciences, Gomal University, Dera Ismail Khan 29220, Pakistan; mtansar5@gmail.com
2    Zoology Department, College of Science, King Saud University, P.O. Box 2455, Riyadh 11451, Saudi Arabia; wadaan@ksu.edu.sa (M.A.W.); fmuhammad@ksu.edu.sa (M.F.K.); almbaabbad@ksu.edu.sa (A.B.)
3    National Center of Excellence in Physical Chemistry, University of Peshawar, Peshawar 25120, Pakistan; drfarooq@uop.edu.pk
4    Department of Biochemistry, KMU Institute of Dental Sciences, Kohat 26000, Pakistan
5    Department of Chemistry, Kohat University of Sciences & Technology, Kohat 26000, Pakistan; drijaz_chem@yahoo.com
6    College of Professional Studies, Northeastern University, Boston, MA 02115, USA; ziaullah.z@northeastern.edu
*    Correspondence: hidayatktk@gu.edu.pk (H.U.); drfoziazeb@yahoo.com (F.F.)

**Abstract:** Accumulation of heavy metals in soil and vegetables is presently a challenging environmental concern worldwide. The present study was designed to elucidate heavy metals contamination of vegetables irrigated with domestic wastewater and associated health risks. The study area comprises three zones: Kot Addu, Alipur, and Muzaffargarh. A total of 153 samples of wastewater, topsoil, and vegetables were analyzed for physicochemical parameters and concentration levels of eight metal elements (Cu, Fe, Zn, Mn, Pb, Cd, Ni, and Cr) determined through analytical procedures. The outcome of the present investigation reveals that heavy metal concentrations in wastewater, soil, and vegetables irrigated with wastewater were slightly higher than the WHO-suggested limit. The heavy metals concentration observed in vegetables irrigated with wastewater can be ranked in order of Ni > Mn > Cr > Pb > Cu > Fe > Zn > Cd. Transfer factor (TF), daily ingestion of metals (DIM), and health risk index (HRI) were calculated. Spinach exhibited higher values of transfer factor than cabbage, cauliflower, and radish, which were followed by tinda and carrot. Minimum values of HRI were observed for Cr (0.0109) in almost all of the vegetables ingested by adults and children. Cabbage exhibited higher values of HRI for Pb (4.0656) in adults, followed by cadmium (HRI = 2.993). Minimum values of HRI were calculated for Cd (0.0115; child). Cauliflower exhibited higher values of HRI (5.2768) for Pb in children. Pb, HRI values (4.5902) were observed in adults living in Kot Addu. The results exhibited similar trends of HRI in adults and children living in Muzaffargarh and Alipur.

**Keywords:** heavy metals; vegetables; metals intake; health risk; wastewater-irrigation

## 1. Introduction

Though often overlooked, water is an imperative gift of nature for numerous lives that depend on it. All the activities of life are performed by water. Only a small fraction (<1%) of fresh water is available for terrestrial ecosystems [1–3]. Approximately 70% of the globally available freshwater is consumed by agricultural activities [4]. Currently, all over the world, the scarcity of water on land is due to numerous factors, including rapid population growth, uneven distribution patterns, abrupt climatic variations, and massive contamination by industrial and agricultural activities [5]. Owing to the shortage of freshwater around the world, most people are becoming more reliant on groundwater, which is characteristically expensive and also of poor quality because of high SAR (sodium adsorption ratio), higher

cations exchange capacity, excessive sodium salts of carbonate, bicarbonate, and heavy metals. However, domestic and industrial sewage waste has become an alternative source of water for irrigation intentions. It has been estimated that the annual world wastewater production in arid and semi-arid regions is 1500 km$^3$, and about 20 million hectors (7% of the total irrigated) of land is irrigated with untreated wastewater for crop production [6,7]. The wastewater is the farmers' second choice for using this water in agricultural drives when there is a freshwater shortage. The quality of wastewater varies in terms of chemical composition and depends on sources of effluent production [8]. Recently, the recycling of wastewater in agriculture has become a widespread practice in regions located in dry arid zones, where scarcity of water is more pronounced [9]. The production of healthy food is a major concern all over the ecosphere [10]. The main source of income in Pakistan's remote villages, particularly in Sothern Punjab, is agriculture and growing vegetable crops commercially. Vegetables and fruits are important constraints for maintaining good health. These are sources of vitamins and important nutrients and support the body's metabolism. WHO suggested that daily intake of vegetables and fruits should be not less than 400 g per person in terms of acquiring numerous micro and macro-nutrients necessary for balanced health stability [11,12]. Unfortunately, in Pakistan, the average daily intake of vegetables is 130 g, which is 66% lower than the defined limit per day by WHO [13]. Besides the importance of vegetables, no information is available about the preferences of particular vegetables by local people and the magnitude of consumption of vegetables at international as well as national and regional levels. In underdeveloped countries, people consume vegetables as a source of food [14]. Humans grow vegetables in sewage-treated land, causing the heavy metals to enter the body. Long-term practices of irrigation through wastewater may lead to the accumulation of heavy metals in various crop plants and agricultural soil. Most vegetable crops are facing the severe problem of heavy metal pollution all over the world. The impacts of heavy metal pollution on vegetable growth and associated health risks are more pronounced in the areas close to industrial sites and the crops irrigated with domestic and industrial effluents [15]. Heavy metals having a non-biodegradable nature, usually known as trace elements, are essential in water sources [16]. Toxic concentrations of all trace metals, both necessary and unnecessary, affect biological processes, destroy cell membranes, and affect the three-dimensional structure of enzymes [16]. Some elements like copper, zinc, and selenium are important for maintaining human metabolic activities at the cellular level in various organs [17]. Arsenic, lead, cadmium, chromium, cobalt, and mercury are assumed to be potentially hazardous and exhibit negative health consequences, if consumed for a long time and ultimately lead to cancerous diseases [18]. Important constituents of vitamins, protein, iron, calcium, and other nutrients are directly obtained from the vegetables [19]. A higher concentration of Zn can cause shock, frits, vomiting, and forehead irritation [20]. The composition of white and red blood cells and severe digestive problems are associated with low concentrations of arsenic, while increased amounts result in black spots on the soles of the feet and brownish skin. The essential activities of enzymes are stimulated when the concentration level of Ni is low, and Ni has a detrimental effect in excessive amounts [21]. The level of health risks posed by wastewater with heavy metals was quantified by the application of different indices, including the transfer factor (TF), daily intake of metals (DIM), and health risk index (HRI). Little information is available about the impacts of heavy metals contamination on vegetables irrigated by sewage wastewater to common people of society. To recognize the health risks caused by vegetables contaminated with excessive concentrations of heavy metals, suitable information is very important. Reference [3] made studies to examine the influence of heavy metals toxicity in vegetable crops: broad bean (*Vacia faba*), durum wheat (*Triticum turgidum*), soft wheat (*Triticum æstivum*), oat (*Avena sativa*), nettle (*Urtica dioica*), broadleaf plantain (*Plantago major*), alfalfa (*Medicago sativa*), and mallow (*Malva sylvestris*), irrigated waste wastewater in Marrakech city, Morocco. Reference [4] selected the three vegetables: lettuce, cabbage, and tomato, for elucidation of the health risks of heavy metals from vegetables grown on soil irrigated with untreated and

treated wastewater in Arba Minch, Ethiopia. Reference [22] conducted research work in India to explore the impact of heavy metals accumulation in vegetables including coriander, onion, and tomato. Reference [23] had investigated the impacts of heavy metal pollution on Ethiopian agriculture. Reference [24] revealed the role of wastewater use for the irrigation of vegetation crops and associated health risks. They had selected only two species of vegetables, namely cauliflower and cabbage. However, the data matrix related to heavy metals pollution in underground water, soil, and vegetables had not yet been examined in the area under consideration (Kot Addu, Muzaffargarh and Alipur), Pakistan. Therefore, the current investigations were made to elucidate the heavy metals contamination of vegetables irrigated with domestic sewage wastewater and associated health risks.

## 2. Materials and Methods

### 2.1. Study Area

The study area comprises three zones (Figure 1), namely classified as Kot Addu (30.4685° N, 70.9606° E), Alipur (29.3817° N, 70.9131° E), and Muzaffargarh (30.0736° N, 71.1805° E), located along elevation gradient from 103 m (a.s.l) to 125 m (a.s.l). The study was designed to cover the 8249 Km$^2$ area of district Muzaffargarh, part of the southern Punjab province of Pakistan. The area under consideration is classified as subtropical dunes. The climate is very hot, with a harsh dry season in summers with maximum values recorded for temperature at approximately 54 °C (129 °F) and winter with minimum values recorded for temperature at approximately −1 °C (30 °F). Rainfall concentrates from late July to August, and the area receives an average annual rainfall of 127 mm (5.0 in). Wind erosion is common, characterized by dust storms. According to the census 2017 report, a total of 4,322,009 persons harbor in the area.

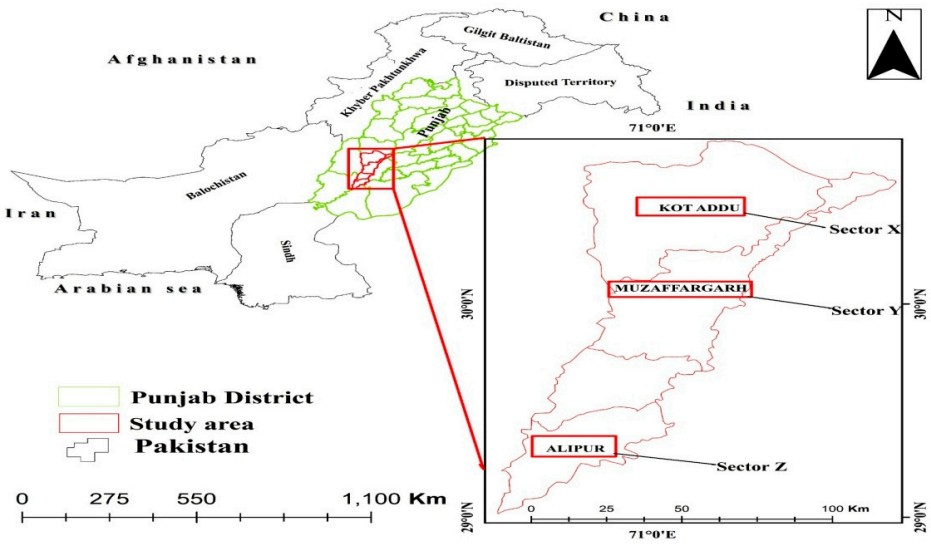

**Figure 1.** Showing Study area for Sampling: Kot Addu, Muzafargarh, and Alipur.

The study area has been classified into three zones: (i) Kot Addu designated as sector X, (ii) Muzaffargarh designated as sector Y and (iii) Alipur designated as sector Z. In contrast, sector Z was used to collect fresh water, irrigated soil, and vegetables. Samplings were performed during 2021 and 2022. Sectors X and Y are separated by 60 km, which is the longest distance, and 9 km, which is the shortest distance between subsectors of each prescribed sector. From the examining areas, tests of sewage water, soil, and vegetables were gathered. Sector Z, on the other hand, was used to collect freshwater samples along with their irrigated soil and vegetables. Standard Recommended Methods (SRM) were used for each sampling and analysis.

## 2.2. Field Sampling

Sampling of wastewater was carried out from October 2021 to April 2022, from various disposal sectors and explored to determine the heavy metals contents. Different sampling procedures were applied to evaluate the quality of water used for irrigation of vegetable crops [5]. Sewage wastewater samples were collected in 500 mL plastic bottles from all the sources present in the area under consideration. All the bottles were washed with 5% nitric acid, then treated with deionized water and cleaned twice. A triplicate sample was taken from each sampling site for accuracy. In the study area (sectors X, Y, and Z), composite soil samples were collected from topsoil at 0 to 30 cm depths and stored in plastic bags. Composite samples of broad-leaved vegetables including *Spinacia oleracea* (spinach), *Brassica oleracea var. capitata* (cabbages), *Brassica oleracea var. botrytis* (cauliflower), *Raphanus sativus* (radish), *Brassica rapa* subsp. rapa (turnip), *Praecitrullus fistulosus* (tinda), *Daucus carota* (carrots), *Lactuca sativa* (lettuce), and *Colocasia esculenta* (Colocasia) were collected from the study area irrigated with sewage and partially sewage water. The vegetable samples were cleaned with distilled water to eliminate dust and other pollutant matter, and for specimen drying purposes, 105 °C temperatures were fixed in the oven for 12 h. The dried samples were then pulverized, homogenized, and transferred into clean sample bottles. For analysis, only the edible section of the vegetable was used.

## 2.3. Chemical Analysis

### 2.3.1. Determination of Biochemical Composition and Concentration of Trace Metals in Wastewater Samples

The collected water samples were used to determine the pH, electrical conductivity of wastewater, and total dissolved solids (TDS). pH was determined by using a digital pH meter (Jenway 3510, Tokyo, Japan). For additional analysis of the water sample, SRM was used [5,25]. The cation exchange capacity of collected water samples was determined by using a digital EC meter. KCl standard solution was used for calibration. The electrode bulb was washed with deionized water after analyzing each sample [26,27]. Utilizing a movable meter (Jenway 4510 Conductivity/TDS Meter; 230 VAC/UK), the recommended methods for estimating TDS for samples were used [28,29]. The samples of sewage water and unadulterated water were gathered over more than seven days. The sewage and new water processing was finished by blending 50 mL of the samples with 10 mL moved nitric corrosive in a particular sample volume ($HNO_3$). The digesting process was prolonged in a water bath for up to one hour at eighty degrees Celsius and then filtered. The filtrate was diluted after being filtered using Whatman (no-42) filter sheets and 0.05 L deionized water. We utilized a Perkin Elmer Atomic absorption spectrophotometer (AAS) model 7000 to distinguish the measures of heavy metals such as Cr, Pb, Cd, Cu, Fe, Ni, Zn, and Mn. To guarantee accurate results, standard solutions of each of these metals were employed.

### 2.3.2. Determination of Biochemical Composition and Concentration of Trace Metals in Soil Samples

The soil samples' pH, EC, and organic matter (OM) were calculated. SRM was followed to perform analysis. The nature of edaphic features, soil chemical reactions (pH), and nutrient cycling between the soils and vegetables are the important dimensions to evaluate the sewage quality in terms of heavy metals contamination. Soil sample extracts were prepared and then pH was determined by using a digital pH meter, following the methods of [29]. Soil electrical conductivity was determined using a digital EC meter by following standard procedure [25,30]. The amount of soil organic matter content was determined using the Walkley–Black method [27,31]. Heavy metal testing was also conducted on the subsurface soil. A 500 mL dried soil sample was mixed with 15 mL of a 5:1:1 solution of $HNO_3$, $H_2SO_4$, and $HClO_4$ overnight. At two separate temperatures, the first digestion took place for 30 min at 50 °C. The second digestion was carried out for 180 min at 120 °C. A VELP Scientific digester was employed for indigestion, and it was kept running until transparent solutions were produced. The Whatman filter paper was used for the filtering

process (no-42). A total of 50 mL of deionized water was used to dilute the filtrate [13,17]. Perkin Elmer's Atomic absorption spectrophotometer (AAS) model 7000 was used for further analysis.

2.3.3. Determination of Biochemical Composition and Concentration of Trace Metals in Vegetable Samples

The vegetable species under consideration after sampling were subjected to biochemical analysis to estimate the heavy metals (Cr, Pb, Cd, Cu, Fe, Ni, Zn, and Mn) contents by following the standard procedure [27]. After sampling the vegetable specimens from the study area, each specimen was dried for 24 h and completely cleaned with distilled water followed by 1% HCl to remove any unwanted particles. Only a part of the vegetables was used for chemical analysis, following standard procedure [27]. The dried material was crushed into a fine powder and stored in tidy containers that were properly sealed. With 50 mL of deionized water, the filtrate was made more weakened [5]. Perkin Elmer's Atomic absorption spectrophotometer (AAS) model 7000 was used for further analysis.

*2.4. Health Risk Assessment*

How harmful contaminated vegetables are to one's health is determined by their hazard quotient. The hazard quotient describes the correlation between the reference dosage and the computed dose. The general public views it as safe if the ratio is less than one. The population will be in grave danger if the value is greater than one or the same as the esteem. The researchers were proven to be accurate while using this method [31]. The equation reads as follows:

$$HQ = (W_{plant}) \times (M_{plant})/R_fD \times B$$

$(W_{plant})$ = Dry weight of a contaminated piece of the plant that is consumed in mg/d.
$(M_{plant})$ = Heavy metal substance that is tracked down in vegetables in mg/kg.
$R_fD$ = Typical grouping of metal in food $(mgd^{-1})$.
B = The typical mass of the body is determined in kg.
Adult = 55.9 kg, Child = 32.7 kg.

The conventional values and RfD values for harmful excess heavy metals were developed by the Department of Food and Rural Affairs, the Department of the Environment, and the Institute of Integrated Risk Information.

*2.5. Determination of Transfer Factor (TF)*

The transfer factor refers to the estimation of the limit of easily heavy metals that can move from soil to vegetative parts of crops. If the value of the transfer factor is greater than one, then the plants showed a strong ability to absorb metals from agricultural soil. The TF was determined by using Equation [32].

$$TF = C \ plant/CAS$$

where C plant = Heavy metals in edible fragments of vegetables
CAS = HMs concentration within agricultural soil (mg/Kg dry weight)

*2.6. Daily Dietary Index (DDI)*

The daily consumption of dietary diversity indices was determined by the following [33].

*2.7. Health Risk Index (HRI)*

To determine the "chronic health risks" HRIs of heavy metals contents were measured by using the following Equation [15]. The health risk was determined by utilizing DIM and oral dose. An HRI value of less than one (HRI < 1) is suggested to be safe for health [24].
HRI was determined as:

$$HRI = DIM/RfD$$

## 3. Statistical Analysis

The data obtained from the lab experiment were statistically analyzed by IBM SPSS software (Version 26) computer program. Analysis of variance (ANOVA) was applied for the exploration of the impact of heavy metals on the quality of vegetables to describe pictorial irrigation water situations [34]. The presence of heavy metals and health risks were statistically evaluated [35].

## 4. Results and Discussion

### 4.1. Exploration of Heavy Metals Concentration in Wastewater

The physicochemical attributes of the wastewater collected from the three different sectors (X, Y, and Z) are presented (Table 1). The result of the analysis of variance (ANOVA) described that there were pronounced variations among pH levels of wastewater samples collected from three different sectors (X, Y, Z). The pH values ranged from 8.37 to $7.80 \pm 0.28$, 8.70 to $7.40 \pm 0.65$, 7.25 to $7.04 \pm 0.11$, respectively ($F = 26.96$; $p < 0.000$). The WHO acceptable pH limit for water is 6.5 to 8.8. However, wastewater comprises carbonates of sodium and calcium along with sodium chlorides and sodium Sulphate. Due to the presence of these salts, the pH of wastewater was slightly higher and exhibited a more alkaline nature than the pH of fresh water. The trends found in the chemical reaction (pH) of wastewater were also reported by [36]; they elucidated the physiochemical attributes of wastewater in Bhakkar. The result (Table 1) reveals that there were significant variations exist in terms of EC among the three sectors ($F$ & $p$-value = 474.06***). EC values of wastewater ranged from 1740 to $1680 \pm 30.00$ μS/cm, 1710 to $1640 \pm 35.00$ μSc/m, and 1335 to $1298 \pm 18.50$ μS/cm, respectively. Similar findings were also reported by [15]. The result (Table 1) suggested that wastewater samples collected from the area under consideration resources were saline. In most of the analyzed wastewater samples, EC values were above the permissible limit (1400 μS/cm) guided by [37]. The exceeded values of EC in wastewater are not surprising, because numerous researchers also reported similar findings [38]. In the wastewater used for irrigation, heavy metals Pb, Cr, Cd, Cu, Zn, Ni, Fe, and Mn had a range of 0.28 to $0.45 \pm 0.09$, $F$ & $p$-values = 26.68***; 0.77 to $0.98 \pm 0.11$, $F$ & $p$-values = 100.78***; 0.64 to $0.96 \pm 0.16$, $F$ & $p$-values 69.72***; 0.32 to $0.48 \pm 0.08$, 0.14 to $0.42 \pm 0.14$, $F$ & $p$-values = 29.74***; 0.29 to $0.47 \pm 0.09$, $F$ & $p$-values = 11.25*** 8.14 to $10.35 \pm 1.11$, $F$ & $p$-values = 83.82***; 0.3 to $0.52 \pm 0.11$, $F$ & $p$-values= 7.32*** (Table 1) in sector X, respectively, and exhibited significant variation for all the variables (Table 1). Similarly, the heavy metals concentrations observed in wastewater collected from sector Y showed significant differences (Table 1) and ranged from 0.22 to $0.45 \pm 0.12$, 0.68 to $0.96 \pm 0.14$, 0.52 to $0.77 \pm 0.13$, 0.35 to $0.40 \pm 0.03$, 0.20 to $0.27 \pm 0.04$, 0.25 to $0.37 \pm 0.06$, 9.65 to $10.42 \pm 0.39$, 0.31 to $0.43 \pm 0.06$, respectively. In the freshwater collected from sector Z used for irrigation, heavy metals Pb, Cr, Cd, Cu, Zn, Ni, Fe, and Mn had range of 0.08 to $0.10 \pm 0.01$, 0.07 to $0.15 \pm 0.04$, 0.06 to $0.08 \pm 0.01$, 0.14 to $0.22 \pm 0.04$, 0.07 to $0.1 \pm 0.02$, 0.10 to $0.22 \pm 0.06$, 4.78 to $5.9 \pm 0.56$, 0.17 to $0.32 \pm 0.08$, respectively (Table 1), similar heavy metals contamination constraints were reported by [10]. The magnitude of heavy metals contamination in wastewater was ranked in order of Fe> Cr > Mn > Cd > Ni > Cu > Zn > Pb. The results demonstrated that all the heavy metal contents overflowed from the allowed limits good for health suggested by WHO, used for irrigation drives.

**Table 1.** Physiochemical attributes of wastewater and heavy metals concentration. Samples were collected from three sectors (X, Y, and Z).

| Variables | pH | EC (μS/cm) | TDS (mg/L) | Pb (mg/L) | Cr (mg/L) | Cd (mg/L) | Cu (mg/L) | Zn (mg/L) | Ni (mg/L) | Fe (mg/L) | Mn (mg/L) |
|---|---|---|---|---|---|---|---|---|---|---|---|
| Sector X | $8.07 \pm 0.28$ | $1710 \pm 30.00$ | $1120.5 \pm 24.50$ | $0.36 \pm 0.09$ | $0.87 \pm 0.11$ | $0.8 \pm 0.16$ | $0.4 \pm 0.08$ | $0.28 \pm 0.14$ | $0.38 \pm 0.09$ | $9.245 \pm 1.11$ | $0.41 \pm 0.11$ |
| Min–Max | 7.80–8.35 | 1680.00–1740.00 | 1096.00–1145.00 | 0.28–0.45 | 0.77–0.98 | 0.64–0.96 | 0.32–0.48 | 0.14–0.42 | 0.29–0.47 | 8.14–10.35 | 0.30–0.52 |

**Table 1.** *Cont.*

| Variables | pH | EC (µS/cm) | TDS (mg/L) | Pb (mg/L) | Cr (mg/L) | Cd (mg/L) | Cu (mg/L) | Zn (mg/L) | Ni (mg/L) | Fe (mg/L) | Mn (mg/L) |
|---|---|---|---|---|---|---|---|---|---|---|---|
| Sector Y | 8.05 ± 0.65 | 1675 ± 35.00 | 1090 ± 30.00 | 0.33 ± 0.12 | 0.82 ± 0.14 | 0.64 ± 0.13 | 0.37 ± 0.03 | 0.23 ± 0.04 | 0.31 ± 0.06 | 10.03 ± 0.39 | 0.37 ± 0.06 |
| Min–Max | 7.40–8.70 | 1640–1710 | 1060–1120 | 0.22–0.45 | 0.68–0.96 | 0.52–0.77 | 0.35–040 | 0.20–0.27 | 0.25–0.37 | 9.65–10.42 | 0.31–0.43 |
| Sector Z | 7.145 ± 0.11 | 1316.5 ± 18.50 | 695 ± 15.00 | 0.09 ± 0.01 | 0.11 ± 0.04 | 0.07 ± 0.01 | 0.18 ± 0.04 | 0.08 ± 0.02 | 0.16 ± 0.06 | 5.34 ± 0.56 | 0.24 ± 0.08 |
| Min–Max | 7.04–7.25 | 1298–1335 | 680–710 | 0.08–0.10 | 0.07–0.15 | 0.06–0.08 | 0.14–0.22 | 0.07–0.10 | 0.10–0.22 | 4.78–5.90 | 0.17–0.32 |
| F & p-value | 26.96 *** | 474.06 *** | 596.23 *** | 26.68 *** | 100.78 *** | 69.72 *** | 29.74 *** | 6.45 * | 11.25 *** | 83.82 *** | 7.32 *** |
| WHO, 2016 | 6.5–8.8 | 1400 | 1000 | 0.1 | 0.1 | 0.01 | 0.2 | 2 | 0.2 | 5 | 0.2 |

* $p < 0.05$, *** $p < 0.001$; StD = values are expressed in ±; (i) Kot Addu designated as sector X, (ii) Muzaffargarh designated as sector Y, and (iii) Alipur designated as sector Z.

## 4.2. Exploration of Heavy Metals Concentration in Soil Irrigated with Wastewater

Biochemical properties of soil samples collected from the study area irrigated with sewage wastewater were analyzed to determine the level of heavy metals' heterogeneity. Results of the ANOVA (Table 2) revealed that there were no significant differences observed among the three sectors for the pH of soil irrigated with wastewater ($F$ & $p$-values= 0.29[NS]). pH ranged from 8.11 to 8.18. The values of soil chemical reactions (pH) were within the range as suggested by WHO. The slightly alkaline nature of wastewater-irrigated soil is because sewage waste is comprised of sufficient content of magnesium, calcium, and bicarbonates. The EC of soil samples collected from the area irrigated with wastewater ranged from 457.50 EC (µS/cm) to 466.25 EC (µS/cm) and exhibited significant differences ($F$ & $p$-values = 50.49***). The results demonstrated that the EC (µS/cm) of the wastewater was not below the WHO's limits (1400 µS/cm), hence the EC of wastewater-irrigated soil was not acceptable for irrigation drives. These findings in terms of electrical conductivity of soil were similar to the outcomes of [39]. The soil was enriched with decomposable ingredients, resulting in considerable content of organic matter (%) ranging from 1.375 to 1.4725. Our results are in line parallel with the findings reported by [26], who explored the organic matter of soil in the D.G. Khan District. Heavy metals Pb, Cr, Cd, Cu, Zn, Ni, Fe, and Mn in soil irrigated with wastewater ranged from 74.50, 63.00, 3.20, 35.40, 31.50, 78.00, 220.00, and 310.00 mg/Kg in sector X, 70.75, 56.00, 3.35, 38.00, 33.00, 85.50, 230.00, and 365.00 values were observed in sector Y, respectively. Similarly, 46.25, 34.25, 1.25, 16.75, 15.75, 36.50, 192.50, and 134.25 mg/Kg were observed in sector Z, accordingly. The results followed the permissible limits for heavy metals described by WHO. The heavy metals contamination in soil irrigated with sewage wastewater can rank in the following order: Mn > Fe > Ni > Pb > Cr > Cu > Zn > Cd (Table 2).

**Table 2.** Heavy metals accumulation in soil irrigated with wastewater. Samples were collected from three sectors (X, Y, and Z).

| Variables | pH | EC (µS/cm) | O.M (%) | Pb (mg/L) | Cr (mg/L) | Cd (mg/L) | Cu (mg/L) | Zn (mg/L) | Ni (mg/L) | Fe (mg/L) | Mn (mg/L) |
|---|---|---|---|---|---|---|---|---|---|---|---|
| Sector X | 8.18 ± 0.28 | 457.5 ± 22.50 | 1.37 ± 0.02 | 74.5 ± 6.50 | 63 ± 6.00 | 3.2 ± 0.30 | 35.4 ± 1.60 | 31.5 ± 2.50 | 78 ± 6.00 | 220 ± 25.00 | 310 ± 30.00 |
| Min–Max | 7.90–8.47 | 335.00–480.00 | 1.35–1.40 | 68.00–81.00 | 57.00–69.00 | 2.90–3.50 | 33.80–37.00 | 29.00–34.00 | 72.00–84.00 | 195.00–245.00 | 280.00–340.00 |
| Sector Y | 8.11 ± 0.24 | 466.25 ± 14.98 | 1.47 ± 0.12 | 70.75 ± 7.98 | 56 ± 5.51 | 3.35 ± 0.64 | 38 ± 4.98 | 33 ± 6.01 | 85.5 ± 7.49 | 230 ± 39.98 | 365 ± 45.03 |
| Min–Max | 7.80–8.30 | 450.00–480.00 | 1.35–1.60 | 62.00–78.00 | 49.00–60.00 | 2.60–3.90 | 32.00–42.00 | 27.00–39.00 | 78.00–93.00 | 180–260 | 310–400 |
| Sector Z | 8.07 ± 0.35 | 303.75 ± 32.49 | 0.85 ± 0.44 | 46.2 ± 8.49 | 34.25 ± 3.99 | 1.25 ± 0.39 | 16.75 ± 4.49 | 15.75 ± 2.98 | 36.5 ± 5.49 | 192.5 ± 22.49 | 134.25 ± 12.98 |
| Min–Max | 7.73–8.07 | 275.00–303.75 | 0.40–0.85 | 38.00 | 46.25 | 31.00–34.25 | 0.90–1.25 | 13.00–16.75 | 13.00–15.75 | 30.00–36.50 | 175.00–192.50 |

**Table 2.** *Cont.*

| Variables | pH | EC (μS/cm) | O.M (%) | Pb (mg/L) | Cr (mg/L) | Cd (mg/L) | Cu (mg/L) | Zn (mg/L) | Ni (mg/L) | Fe (mg/L) | Mn (mg/L) |
|---|---|---|---|---|---|---|---|---|---|---|---|
| *F* & *p*-value | 0.29 NS | 50.49 *** | 10.60 *** | 17.41 *** | 62.34 *** | 49.31 *** | 32.22 *** | 22.65 *** | 78.85 *** | 2.36 NS | 82.82 *** |
| WHO, 2007/2016 | -- | -- | -- | 60 | 100 | 1 | 100 | 200 | 67.9 | 150 | 500 |

*** $p < 0.001$; NS: Non-significant; StD = values are expressed in $\pm$; (i) Kot Addu designated as sector X, (ii) Muzaffargarh designated as sector Y, and (iii) Alipur designated as sector Z.

### 4.3. Exploration of Heavy Metals in Vegetables Irrigated with Wastewater

In the present investigation, determination of different levels of heavy metals accumulated in vegetables and crop plants, irrigated with municipal sewage waste, industrial waste was the major concern and compared with the permissible limits set by WHO/FAO. Heavy metals Cu, Fe, Zn, Mn, Pb, Cd, Ni, and Cr in vegetables irrigated with wastewater from sector X also showed significant differences in almost all the variables (Table 3), and all the metals ranged from 9 to 29, (*F* & *p*-values = 20.48***) 6.3 to 26.75, (*F* & *p*-values 4.89**); 7.50 to 22.30, (*F* & *p*-values = 4.46**); 10.30–38.40, (*F* & *p*-values = 19.67***); 6.00 to 35.00, (*F* & *p*-values = 10.57***); 14.30 to 42.50 (*F* & *p*-values = 13.26***); 14.30 to 28.20, (*F* & *p*-values = 3.36**), and 4.30 to 38.00 mg/Kg (*F* & *p*-values = 1.59*) and those from sector Y wastewater-irrigated vegetables 11.30 to 23.40, 5.70 to 26.40, 6.78 to 17.75, 11.45 to 41.38, 7.40 to 38.00, 2.9 to 8.48, 14.90 to 48.84 and 10.47 to 34.00 mg/Kg, respectively. Cu, Fe, Zn, Mn, Pb, Cd, Ni, and Cr in vegetables irrigated with fresh water from the sector ranged from 6.95 to 15.80, 2.45 to 48.95, 2.96 to 14.32, 3.40 to 22.45, 4.31 to 10.35, 0.90 to 3.70, 4.38 to 29.92, and 4.37 to 27.40 mg/Kg, respectively. The result (Table 3) reveals that heavy metal elements, Mn, Pb, Cd, Ni, and Cr had overriding values in vegetables and exhibited exceeded levels suggested by the World Health Organization (WHO). These results are similar to the findings of [10]. The magnitude of contamination of vegetables irrigated with wastewater can be ranked: Ni > Mn > Cr > Fe> Pb > Cu > Zn > Cd. It can be concluded that nickel, manganese, and chromium (mg/Kg) contents were higher in all the vegetables under consideration. The present investigation reveals that if local farmers continue the practice of using sewage wastewater and untreated municipal industrial waste for the irrigation of vegetables may lead to the accumulation of heavy metals, particularly Cd, Cr, and Pb may occur from water to plants and livestock. These results are congruent with the findings of [40,41]. In the present research, nickel concentration was found in an excessive amount in vegetables irrigated with sewage wastewater, soil, and water samples subjected to chemical analysis (Table 3). This is because the study area is a dry subtropical sand dunes desert zone facing the regular windblown dust storm derived from the weathering of rocks and soil from the Suleman Mountain range, Dera Ghazi Khan.

The presence of nickel in the air also accumulates from the combustion of coal, fuel, petrol, and diesel by vehicles and thermal power plants occurring in nearby areas of the study area. Similar findings in terms of nickel sources were also reported by [13]. The higher level of nickel in vegetables and other fruit stuff poses severe health risks (Table 3), and these findings were in parallel to the results of [5]. In the soil, nickel accumulation is sustained by earthworms and micro-organisms [42]. Many other environmental factors also cause the accumulation of excessive heavy metals [43]. Another cause of the higher content of nickel in vegetation crop plants is due to nickel being preferred by all angiospermic plant species as it plays an imperative role in the conversion of urea into ammonium [43]. Chromium exhibited an excessive level in the present investigation (Table 3) than the suitable amount for health suggested by WHO. Many industries have been established in the area under consideration and producing chromium pollution in water and soil. The effluent discharge from these industries contains much Cr content in its composition [26]. Besides these, Zinc, Pb, Cd, and Mn should be further scrutinized in derived products from the study crops and livestock, particularly meat, milk, and leafy vegetables to determine considerable food

safety problems. Besides these, wastewater reuse may cause environmental risks along with health diseases in humans [36].

**Table 3.** Heavy metals contents in vegetables irrigated with wastewater. Samples were collected from three sectors (X, Y, and Z).

| Variables | Pb (mg/L) | Cr (mg/L) | Cd (mg/L) | Cu (mg/L) | Zn (mg/L) | Ni (mg/L) | Fe (mg/L) | Mn (mg/L) |
|---|---|---|---|---|---|---|---|---|
| Sector X Min–Max | $20.5 \pm 14.50$ | $21.15 \pm 16.85$ | $4.53 \pm 3.23$ | $19 \pm 10.00$ | $14.9 \pm 7.40$ | $28.4 \pm 14.10$ | $16.53 \pm 10.23$ | $24.35 \pm 14.05$ |
| | 6.00–35.00 | 4.30–38.00 | 14.30–42.50 | 9.00–29.00 | 7.50–22.30 | 14.30–28.20 | 6.30–20.45 | 10.30–38.40 |
| Sector Y Min–Max | $22.70 \pm 6.05$ | $22.24 \pm 10.35$ | $5.69 \pm 5.49$ | $17.35 \pm 14.97$ | $12.27 \pm 15.30$ | $31.87 \pm 2.79$ | $16.05 \pm 16.97$ | $26.42 \pm 11.77$ |
| | 7.40–38.00 | 10.47–34.00 | 2.90–8.48 | 11.30–23.40 | 6.78–17.75 | 14.90–48.84 | 5.70–26.40 | 11.45–41.38 |
| Sector Z Min–Max | $7.33 \pm 4.43$ | $15.89 \pm 23.25$ | $2.35 \pm 5.68$ | $11.38 \pm 9.53$ | $8.64 \pm 3.02$ | $17.15 \pm 1.36$ | $25.7 \pm 12.77$ | $12.93 \pm 11.52$ |
| | 4.31–10.35 | 4.37–27.40 | 0.99–3.70 | 6.95–15.80 | 2.96–14.32 | 4.38–29.92 | 2.45–48.95 | 3.40–22.45 |
| F & *p*-value | 10.57 *** | 1.59 NS | 13.26 *** | 20.48 *** | 4.46 ** | 3.36 ** | 4.89 ** | 19.67 *** |
| WHO, 2007/2016 | 0.3 | 2.3 | 0.2 | 73 | 99 | 0.1 | 425 | 02 |

** $p < 0.01$, *** $p < 0.001$; NS: non-significant; StD = values are expressed in $\pm$; (i) Kot Addu designated as sector X, (ii) Muzaffargarh designated as sector Y, and (iii) Alipur designated as sector Z.

Pearson Correlation Analysis of Heavy Metals in Vegetables Irrigated with Wastewater

Selected species of vegetables irrigated with wastewater were subjected to physio-chemical analysis. The outcomes of the chemical analysis of vegetables were then used for statistical analysis to determine the interrelationship of heavy metals and delimit the major sources contributing to the heavy metals in wastewater used for irrigation and vegetable species accumulate excessive amounts of heavy metals in their vegetative parts. Pearson's correlation was performed by using SPSS software (Version 26) computer program. The results (Table 4) show that if the magnitude of one heavy metal concentration increases in vegetable vegetative parts, then the other concentrations will also increase accordingly and vice versa. The greater "r—value; $p < 0.001$" exhibited a stronger correlation of the two heavy metals to each other. Among the seven types of heavy metals under consideration, Cu exhibited a strong positive correlation with Pb (r = 0.688; $p < 0.001$), and Cu showed a strong positive correlation with Cr (r = 0.605; $p < 0.001$). Similarly, there were highly significant correlations accounted for Zn and Ni (r = 0.638; $p < 0.001$) followed by Zn and Pb (r = 0.458; $p < 0.001$). Besides these, lower values of correlation were observed between Zn and Cd (r = 0.060; $p < 0.002$; Table 4). The results demonstrated that maximum values of the correlation coefficient suggest the common source of similar metal accumulation in vegetables by absorbing wastewater.

**Table 4.** Representing the Pearson's correlation matrix of heavy metal elements in wastewater collected from three different sectors (X, Y, and Z).

| Metals | Cu | Fe | Zn | Mn | Pb | Cd | Ni |
|---|---|---|---|---|---|---|---|
| Fe | 0.028 | | | | | | |
| | 0.889 | | | | | | |
| Zn | 0.275 | 0.154 | | | | | |
| | 0.164 | 0.443 | | | | | |
| Mn | 0.283 | −0.333 | 0.226 | | | | |
| | 0.153 | 0.090 | 0.257 | | | | |
| Pb | 0.688 | 0.120 | 0.458 | 0.081 | | | |
| | 0.000 | 0.552 | 0.016 | 0.687 | | | |
| Cd | 0.334 | -0.108 | 0.060 | 0.566 | 0.049 | | |
| | 0.590 | 0.767 | 0.002 | 0.808 | 0.764 | | |
| Ni | 0.153 | 0.010 | 0.638 | 0.351 | 0.098 | 0.251 | |
| | 0.447 | 0.959 | 0.000 | 0.072 | 0.625 | 0.206 | |
| Cr | 0.605 | 0.070 | 0.172 | 0.154 | 0.469 | 0.171 | −0.064 |
| | 0.001 | 0.728 | 0.390 | 0.443 | 0.014 | 0.394 | 0.753 |

### 4.4. Taxation of Heavy Metal Transfer Factor (HMTF)

Transfer factor (TF) is the quotient that exhibited the potential of plants to absorb the metal contents from the soil, which is enriched with metal pollutants. The result (Table 5) demonstrated the heavy metal transfer factor [43]. Heavy metals associated health risks were determined by transfer factor (TF), consuming the vegetables irrigated with wastewater. The nine vegetables exhibited significantly different transfer factors among the three sector types, as all the values of heavy metals transfer factors vary (Table 5). The F-value indicated a highly significant variance in terms of lead (F = 2.49*; Table 5). The lead (TF) ranged from 0.24 to 0.47 (Table 5) in spinach. Chromium also exhibited significant differences in all three sectors (8.84**). All the vegetables exhibited significant differences. Spinach showed range (0.43 to 0.82), cabbage exhibited (TF) for Cr ranged (1.76 to 2.96), cauliflower (0.41 to 0.61), radish (0.07–0.13), turnip (0.17–0.23), tinda (0.17 to 0.35), carrot (0.37 to 0.64), and lettuce (0.41 to 0.70). Ferric (TF) played an overriding role in overall analysis variance, as much of a significant difference exists (12.99***; Table 5). The magnitude of the effect of sector types can be ranked in terms of difference among the nine types of vegetables that accumulate heavy metals from the soil irrigated by wastewater enriched with heavy metals and showed the potential of transfer factor (Fe = 12.99*** > Cr = 8.84*** > Cu = 8.25*** > Mn = 4.67*** > Zn = 2.82* > Cd = 2.78* > Pb = 2.63*; Table 5). Spinach exhibited higher values of variance in terms of Fe and ranged (0.07–0.11). TF of cabbage was (0.08 to 0.12) for Fe, cauliflower showed range (Fe = 0.05 to 0.06), radish (0.13 to 0.15), tinda (Fe = 0.02 to 0.04), carrot (0.01 to 0.10), and lettuce also exhibited similar values of TF in term of Fe accumulation to carrot (0.01 to 0.11), and Colocasia showed the transfer factor values for Fe accumulation was (0.02 to 0.03; Table 5). Similar findings in terms of transfer factor were reported by [24,26] in broad-leaved perennial vegetables.

**Table 5.** Comparison of transfer factor (TF) of selected heavy metals accumulated in vegetables irrigated with wastewater.

| Vegetables | Sectors | Pb | Cr | Cd | Cu | Zn | Ni | Fe | Mn |
|---|---|---|---|---|---|---|---|---|---|
| Spinach | X | 0.37 | 0.43 | 1.05 | 0.76 | 0.56 | 0.54 | 0.11 | 0.11 |
|  | Y | 0.45 | 0.52 | 1.64 | 0.59 | 0.51 | 0.45 | 0.07 | 0.10 |
|  | Z | 0.24 | 0.82 | 2.00 | 0.94 | 0.91 | 0.90 | 0.07 | 0.16 |
| Cabbage | X | 0.42 | 0.53 | 1.76 | 0.65 | 0.71 | 0.41 | 0.12 | 0.03 |
|  | Y | 0.54 | 0.55 | 1.82 | 0.58 | 0.43 | 0.38 | 0.11 | 0.03 |
|  | Z | 0.18 | 0.80 | 2.96 | 0.62 | 0.19 | 0.12 | 0.08 | 0.09 |
| Cauliflower | X | 0.48 | 0.61 | 0.40 | 0.82 | 0.30 | 0.18 | 0.06 | 0.07 |
|  | Y | 0.47 | 0.61 | 0.81 | 0.62 | 0.35 | 0.17 | 0.05 | 0.06 |
|  | Z | 0.09 | 0.41 | 0.72 | 0.83 | 0.52 | 0.57 | 0.05 | 0.03 |
| Radish | X | 0.38 | 0.07 | 0.74 | 0.42 | 0.38 | 0.40 | 0.15 | 0.04 |
|  | Y | 0.39 | 0.09 | 1.01 | 0.30 | 0.41 | 0.35 | 0.14 | 0.04 |
|  | Z | 0.21 | 0.13 | 1.40 | 0.67 | 0.69 | 0.79 | 0.13 | 0.05 |
| Turnip | X | 0.10 | 0.17 | 1.33 | 0.31 | 0.35 | 0.31 | 0.03 | 0.07 |
|  | Y | 0.16 | 0.24 | 2.48 | 0.34 | 0.30 | 0.20 | 0.06 | 0.11 |
|  | Z | 0.18 | 0.35 | 0.79 | 0.41 | 0.50 | 0.55 | 0.04 | 0.15 |
| Tinda | X | 0.18 | 0.19 | 0.77 | 0.35 | 0.46 | 0.39 | 0.03 | 0.10 |
|  | Y | 0.24 | 0.19 | 0.87 | 0.39 | 0.54 | 0.44 | 0.02 | 0.09 |
|  | Z | 0.22 | 0.39 | 0.98 | 0.53 | 0.81 | 0.67 | 0.03 | 0.17 |
| Carrot | X | 0.20 | 0.37 | 2.30 | 0.49 | 0.24 | 0.28 | 0.09 | 0.10 |
|  | Y | 0.45 | 0.44 | 1.28 | 0.57 | 0.25 | 0.26 | 0.10 | 0.10 |
|  | Z | 0.16 | 0.64 | 1.46 | 0.44 | 0.34 | 0.59 | 0.01 | 0.07 |
| Lettuce | X | 0.15 | 0.41 | 2.11 | 0.34 | 0.46 | 0.48 | 0.11 | 0.12 |
|  | Y | 0.20 | 0.46 | 1.70 | 0.36 | 0.45 | 0.57 | 0.11 | 0.10 |
|  | Z | 0.22 | 0.70 | 1.19 | 0.59 | 0.95 | 0.76 | 0.07 | 0.12 |

**Table 5.** *Cont.*

| Vegetables | Sectors | Pb | Cr | Cd | Cu | Zn | Ni | Fe | Mn |
|---|---|---|---|---|---|---|---|---|---|
| | X | 0.08 | 0.26 | 2.39 | 0.59 | 0.25 | 0.40 | 0.03 | 0.09 |
| Colocasia | Y | 0.10 | 0.32 | 2.53 | 0.60 | 0.21 | 0.38 | 0.03 | 0.08 |
| | Z | 0.17 | 0.70 | 1.43 | 0.68 | 0.25 | 0.82 | 0.02 | 0.08 |
| *F* & *p*-Value | | 2.63 * | 8.84 *** | 2.78 * | 8.25 *** | 2.82 * | 2.79 * | 12.99 *** | 4.67 ** |

* $p < 0.05$, ** $p < 0.01$, *** $p < 0.001$.

### 4.5. Daily Intake of Metals (DIM) and Health Risk Index (HRI)

The result (Table 6) described the daily intake of metals by humans and the health risk index. The present research work was conducted to determine human exposure to various levels of heavy metals contaminated vegetables and to elucidate pollution-related hazards to human health. Vegetables are the major sources of food in underdeveloped countries. In Pakistan, vegetables are grown in soil that is irrigated with wastewater, consequently, people consume higher values of heavy metals daily [44]. However, the metals consumption by vegetables irrigated with underground water is less than those grown in soil irrigated with sewage wastewater. The availability of such secondary data provides information about health risks from metal consumption. The daily intake of metals in Kot Addu (sector X), heavy metals such as Pb, Cr, Cd, Ni, Zn, Cu, Fe, and Mn for adults ranged from 0.0058 to 0.783 and for children ranged from 0.0013 to 0.516. The daily intake of metals (Pb, Cr, Cd, Ni, Zn, Cu, Fe, and Mn) in Muzaffargarh (sector Y) ranged from 0.0092 to 0.989 for adults and children ranging from 0.0096 to 0.0.698. Sector Z exhibited lower values of daily intake of heavy metals for adults and children than sector X and sector Y (Table 6) in terms of values mean across nine vegetable types (spinach, cabbage, cauliflower, radish, turnip, tinda, carrot, lettuce, and colocasia). The health risk index was calculated in terms of vegetable consumption. Spinach grown in sector X (Kot Addu) exhibited maximum values of HRI for Pb (3.541) and is followed by cadmium (1.7836), Ni (1.1148), and Mn (0.1326) for adults. Minimum values of HRI were observed for Cr (0.0094) in sector X (Kot Addu). The results (Table 6) exhibited that there were higher values of HRI (4.070) for Pb observed, taken by children living in Kot Addu followed by cadmium (2.0504). Minimum values of HRI were observed for Cr (0.0109). Cabbage also showed parallel values of HRI for Pb (4.0656) in adults, followed by cadmium (HRI= 2.993; Table 6). Minimum values of HRI were calculated Cd (0.0115; Table 5; Child). Cauliflower exhibited higher values of HRI (5.2768) for Pb in children. Pb HRI values of 4.5902 were observed in adults (sector X; Table 6). The results (Table 6) exhibited similar trends of HRI in adults and children living in sector Y and sector Z. However, the magnitude of HRI for sector Z was less than for sector X and sector Y. HRI values of radish were also higher in children (4.2214). Among the vegetables, turnip (HRI; Pb = 0.9180) and tinda (1.7049) exhibited minimum values HRI for adults, and similar trends were recorded for children living in sector X (Table 6). HRI values for carrots in terms of Cd were 3.9135 in adults, which was maximum than the summit. Among the children living in sector X, higher values of HRI were calculated for cadmium (4.4988). Minimum values of HRI were observed for ferric in children (Table 6). Colocasia exhibited the minimum values of HRI for all heavy metals contaminants in adults and children than other vegetables discussed above cultivated in sector x irrigated with wastewater. The HRI value for lead was 0.7869 in adults, Cr (HRI = 0.0057), and cadmium exhibited maximum values of HRI observed in Colocasia (Table 6). Similarly, copper (HRI = 0.2754) in adults was observed. All the remaining heavy metals showed minimum values of HRI by consuming Colocasia irrigated with wastewater in sector X (Table 6). Mirror reflecting to sector X, trends were attributed by sector Y and sector Z for the heavy metals in terms of daily intake and HIR (Tables 6–8).

**Table 6.** HRI and DIM for adults and children consuming different vegetables of sector X.

| Vegetables | Factor | Lead (Pb) | Chromium (Cr) | Cadmium (Cd) | Copper (Cu) | Zinc (Zn) | Nickel (Ni) | Ferric (Fe) | Manganese (Mn) |
|---|---|---|---|---|---|---|---|---|---|
| **Sector X (Kot Addu)** | | | | | | | | | |
| **Adult** | | | | | | | | | |
| Spinach | DIM | 0.0142 | 0.0142 | 0.0018 | 0.0142 | 0.0092 | 0.0223 | 0.0121 | 0.0186 |
| | HRI | 3.5410 | 0.0094 | 1.7836 | 0.3541 | 0.0306 | 1.1148 | 0.0402 | 0.1326 |
| **Child** | | | | | | | | | |
| Spinach | DIM | 0.0163 | 0.0163 | 0.0021 | 0.0163 | 0.0106 | 0.0256 | 0.0139 | 0.0213 |
| | HRI | 4.070 | 0.0109 | 2.0504 | 0.4071 | 0.0352 | 1.2815 | 0.0462 | 0.1525 |
| **Adult** | | | | | | | | | |
| Cabbage | DIM | 0.0163 | 0.0173 | 0.0030 | 0.0121 | 0.0117 | 0.0172 | 0.0139 | 0.0054 |
| | HRI | 4.0656 | 0.0115 | 2.9902 | 0.3016 | 0.0390 | 0.8577 | 0.0463 | 0.0386 |
| **Child** | | | | | | | | | |
| Cabbage | DIM | 0.0187 | 0.0199 | 0.0034 | 0.0139 | 0.0134 | 0.0197 | 0.0160 | 0.0062 |
| | HRI | 4.6737 | 0.0133 | 3.4374 | 0.3468 | 0.0448 | 0.9860 | 0.0533 | 0.0444 |
| **Adult** | | | | | | | | | |
| Cauliflower | DIM | 0.0184 | 0.0199 | 0.0007 | 0.0152 | 0.0050 | 0.0075 | 0.0081 | 0.0114 |
| | HRI | 4.5902 | 0.0133 | 0.6820 | 0.3803 | 0.0165 | 0.3751 | 0.0269 | 0.0813 |
| **Child** | | | | | | | | | |
| Cauliflower | DIM | 0.0211 | 0.0229 | 0.0008 | 0.0175 | 0.0057 | 0.0086 | 0.0234 | 0.0114 |
| | HRI | 5.2768 | 0.0153 | 0.7840 | 0.4372 | 0.0190 | 0.4312 | 0.0269 | 0.0935 |
| **Adult** | | | | | | | | | |
| Radish | DIM | 0.0147 | 0.0023 | 0.0013 | 0.0079 | 0.0062 | 0.0166 | 0.0163 | 0.0071 |
| | HRI | 3.6722 | 0.0015 | 1.2590 | 0.1967 | 0.0206 | 0.8289 | 0.0542 | 0.0510 |
| **Child** | | | | | | | | | |
| Radish | DIM | 0.0169 | 0.0026 | 0.0014 | 0.0090 | 0.0071 | 0.0191 | 0.0187 | 0.0082 |
| | HRI | 4.2214 | 0.0017 | 1.4473 | 0.2261 | 0.0237 | 0.9528 | 0.0623 | 0.0586 |
| **Adult** | | | | | | | | | |
| Turnip | DIM | 0.0037 | 0.0056 | 0.0023 | 0.0058 | 0.0057 | 0.0131 | 0.0039 | 0.0118 |
| | HRI | 0.9180 | 0.0038 | 2.2558 | 0.1443 | 0.0190 | 0.6531 | 0.0129 | 0.0839 |
| **Child** | | | | | | | | | |
| Turnip | DIM | 0.0042 | 0.0065 | 0.0026 | 0.0066 | 0.0065 | 0.6531 | 0.0045 | 0.0135 |
| | HRI | 1.0554 | 0.0043 | 2.5931 | 0.1658 | 0.0218 | 0.7508 | 0.0149 | 0.0965 |
| **Adult** | | | | | | | | | |
| Tinda | DIM | 0.0068 | 0.0061 | 0.0013 | 0.0066 | 0.0075 | 0.0163 | 0.0033 | 0.0158 |
| | HRI | 1.7049 | 0.0041 | 1.3115 | 0.1639 | 0.0250 | 0.8157 | 0.0110 | 0.1132 |
| **Child** | | | | | | | | | |
| Tinda | DIM | 0.0078 | 0.0071 | 0.0015 | 0.0075 | 0.0086 | 0.0188 | 0.0038 | 0.0182 |
| | HRI | 1.9599 | 0.0047 | 1.5076 | 0.1885 | 0.0287 | 0.9378 | 0.0127 | 0.1301 |
| **Adult** | | | | | | | | | |
| Carrot | DIM | 0.0079 | 0.0120 | 0.0039 | 0.0091 | 0.0039 | 0.0118 | 0.0103 | 0.0166 |
| | HRI | 1.9672 | 0.0080 | 3.9135 | 0.2282 | 0.0131 | 0.5902 | 0.0344 | 0.1188 |
| **Child** | | | | | | | | | |
| Carrot | DIM | 0.0090 | 0.0138 | 0.0045 | 0.0105 | 0.0045 | 0.0136 | 0.0119 | 0.0191 |
| | HRI | 2.2615 | 0.0092 | 4.4988 | 0.2623 | 0.0151 | 0.6784 | 0.0396 | 0.1365 |

**Table 6.** *Cont.*

| Sector X (Kot Addu) | | | | | | | | | |
|---|---|---|---|---|---|---|---|---|---|
| **Adult** | | | | | | | | | |
| **Vegetables** | **Factor** | **Lead (Pb)** | **Chromium (Cr)** | **Cadmium (Cd)** | **Copper (Cu)** | **Zinc (Zn)** | **Nickel (Ni)** | **Ferric (Fe)** | **Manganese (Mn)** |
| **Adult** | | | | | | | | | |
| Lettuce | DIM | 0.0058 | 0.0133 | 0.0102 | 0.0062 | 0.0075 | 0.0199 | 0.0118 | 0.0201 |
| | HRI | 1.4426 | 0.0089 | 0.004 | 0.0072 | 0.0250 | 0.9967 | 0.0393 | 0.1439 |
| **Child** | | | | | | | | | |
| Lettuce | DIM | 0.0066 | 0.0153 | 0.0036 | 0.1561 | 0.0086 | 0.0229 | 0.0136 | 0.0232 |
| | HRI | 1.6584 | 0.0102 | 3.5935 | 0.1794 | 0.0287 | 1.1458 | 0.0452 | 0.1654 |
| **Adult** | | | | | | | | | |
| Colocasia | DIM | 0.0031 | 0.0086 | 0.0047 | 0.0110 | 0.0041 | 0.0165 | 0.0032 | 0.0146 |
| | HRI | 0.7869 | 0.0057 | 4.6737 | 0.2754 | 0.0138 | 0.8236 | 0.0107 | 0.1045 |
| **Child** | | | | | | | | | |
| Colocasia | DIM | 0.0036 | 0.0098 | 4.0656 | 0.0127 | 0.0048 | 0.0189 | 0.0037 | 0.0168 |
| | HRI | 0.9046 | 0.0066 | 0.0110 | 0.3166 | 0.0159 | 0.9468 | 0.0123 | 0.1202 |

**Table 7.** HRI and DIM for adults and children consuming different vegetables of sector Y (Muzaffargarh).

| Sector Y (Muzaffargarh) | | | | | | | | | |
|---|---|---|---|---|---|---|---|---|---|
| **Adult** | | | | | | | | | |
| **Vegetables** | **Factor** | **Cu** | **Fe** | **Zn** | **Mn** | **Pb** | **Cd** | **Ni** | **Cr** |
| Spinach | DIM | 0.0117 | 0.0089 | 0.0088 | 0.0196 | 0.0168 | 0.0029 | 0.0201 | 0.0152 |
| | HRI | 0.2925 | 0.0297 | 0.0294 | 0.1401 | 4.1968 | 2.8853 | 1.0072 | 0.0101 |
| **Child** | | | | | | | | | |
| Spinach | DIM | 0.0134 | 0.0103 | 0.0101 | 0.0226 | 0.0193 | 0.0033 | 0.0232 | 0.0175 |
| | HRI | 0.3362 | 0.0342 | 0.0338 | 0.1611 | 4.8245 | 3.3168 | 1.1579 | 0.0117 |
| **Adult** | | | | | | | | | |
| Cabbage | DIM | 0.0115 | 0.0127 | 0.0075 | 0.006 | 0.0199 | 0.0032 | 0.017 | 0.0163 |
| | HRI | 0.2872 | 0.0425 | 0.0251 | 0.0429 | 4.9837 | 3.2 | 0.8525 | 0.0108 |
| **Child** | | | | | | | | | |
| Cabbage | DIM | 0.0132 | 0.0147 | 0.0087 | 0.0069 | 0.0229 | 0.0037 | 0.0196 | 0.0187 |
| | HRI | 0.3302 | 0.0488 | 0.0288 | 0.0493 | 5.7291 | 3.6787 | 0.98 | 0.0125 |
| **Adult** | | | | | | | | | |
| Cauliflower | DIM | 0.0123 | 0.0066 | 0.0061 | 0.0119 | 0.0173 | 0.0014 | 0.0078 | 0.0178 |
| | HRI | 0.3069 | 0.0221 | 0.0204 | 0.0852 | 4.3279 | 1.4164 | 0.3908 | 0.0119 |
| **Child** | | | | | | | | | |
| Cauliflower | DIM | 0.0141 | 0.0076 | 0.007 | 0.0137 | 0.0199 | 0.0016 | 0.009 | 0.0205 |
| | HRI | 0.3528 | 0.0254 | 0.0234 | 0.0979 | 4.9752 | 1.6283 | 0.4493 | 0.0137 |
| **Adult** | | | | | | | | | |
| Radish | DIM | 0.0059 | 0.017 | 0.0071 | 0.0072 | 0.0143 | 0.0018 | 0.0157 | 0.0027 |
| | HRI | 0.1482 | 0.0567 | 0.0236 | 0.0515 | 3.5804 | 1.7836 | 0.7869 | 0.0018 |
| **Child** | | | | | | | | | |
| Radish | DIM | 0.0068 | 0.0195 | 0.0081 | 0.0083 | 0.0165 | 0.0021 | 0.0181 | 0.0031 |
| | HRI | 0.1704 | 0.0651 | 0.0271 | 0.0592 | 4.1159 | 2.0504 | 0.9046 | 0.0021 |

**Table 7.** *Cont.*

| Sector Y (Muzaffargarh) | | | | | | | | | |
|---|---|---|---|---|---|---|---|---|---|
| | | | | **Adult** | | | | | |
| **Vegetables** | **Factor** | **Cu** | **Fe** | **Zn** | **Mn** | **Pb** | **Cd** | **Ni** | **Cr** |
| | | | | **Adult** | | | | | |
| Turnip | DIM | 0.0068 | 0.0077 | 0.0051 | 0.0217 | 0.006 | 0.0044 | 0.0088 | 0.0069 |
| | HRI | 0.1692 | 0.0257 | 0.017 | 0.1551 | 1.5082 | 4.3542 | 0.4394 | 0.0046 |
| | | | | **Child** | | | | | |
| Turnip | DIM | 0.0078 | 0.0089 | 0.0059 | 0.025 | 0.0069 | 0.005 | 0.0101 | 0.008 |
| | HRI | 0.1945 | 0.0295 | 0.0196 | 0.1782 | 1.7338 | 5.0054 | 0.5051 | 0.0053 |
| | | | | **Adult** | | | | | |
| Tinda | DIM | 0.0077 | 0.003 | 0.0093 | 0.0167 | 0.0088 | 0.0015 | 0.0199 | 0.0055 |
| | HRI | 0.1928 | 0.01 | 0.031 | 0.1191 | 2.1902 | 1.5213 | 0.9954 | 0.0037 |
| | | | | **Child** | | | | | |
| Tinda | DIM | 0.0089 | 0.0034 | 0.0107 | 0.0192 | 0.0101 | 0.0017 | 0.0229 | 0.0063 |
| | HRI | 0.2216 | 0.0115 | 0.0357 | 0.1369 | 2.5178 | 1.7489 | 1.1443 | 0.0042 |
| | | | | **Adult** | | | | | |
| Carrot | DIM | 0.0114 | 0.0123 | 0.0042 | 0.0183 | 0.0167 | 0.0023 | 0.0115 | 0.013 |
| | HRI | 0.2846 | 0.0411 | 0.0142 | 0.1307 | 4.1706 | 2.2558 | 0.5744 | 0.0087 |
| | | | | **Child** | | | | | |
| Carrot | DIM | 0.0131 | 0.0142 | 0.0049 | 0.021 | 0.0192 | 0.0026 | 0.0132 | 0.0149 |
| | HRI | 0.3272 | 0.0472 | 0.0163 | 0.1503 | 4.7943 | 2.5931 | 0.6603 | 0.01 |
| | | | | **Adult** | | | | | |
| Lettuce | DIM | 0.0073 | 0.0138 | 0.0078 | 0.0196 | 0.0075 | 0.003 | 0.0256 | 0.0136 |
| | HRI | 0.1818 | 0.0462 | 0.0261 | 0.1403 | 1.8754 | 2.9902 | 1.2811 | 0.0091 |
| | | | | **Child** | | | | | |
| Lettuce | DIM | 0.0084 | 0.0159 | 0.009 | 0.0226 | 0.0086 | 0.0034 | 0.0295 | 0.0156 |
| | HRI | 0.209 | 0.0531 | 0.03 | 0.1613 | 2.1559 | 3.4374 | 1.4727 | 0.0104 |
| | | | | **Adult** | | | | | |
| Colocasia | DIM | 0.0119 | 0.0041 | 0.0036 | 0.0162 | 0.0039 | 0.0044 | 0.0173 | 0.0093 |
| | HRI | 0.2977 | 0.0138 | 0.0119 | 0.1158 | 0.9705 | 4.4486 | 0.863 | 0.0062 |
| | | | | **Child** | | | | | |
| Colocasia | DIM | 0.0137 | 0.0048 | 0.0041 | 0.0186 | 0.0045 | 0.0051 | 0.0198 | 0.0107 |
| | HRI | 0.3422 | 0.0159 | 0.0136 | 0.1331 | 1.1157 | 5.1139 | 0.992 | 0.0072 |

**Table 8.** HRI and DIM for adults and children consuming different vegetables of sector Z (Alipur).

| Sector Z (Alipur) | | | | | | | | | |
|---|---|---|---|---|---|---|---|---|---|
| | | | | **Adult** | | | | | |
| **Vegetables** | | **Cu** | **Fe** | **Zn** | **Mn** | **Pb** | **Cd** | **Ni** | **Cr** |
| Spinach | DIM | 0.0083 | 0.0069 | 0.0075 | 0.0113 | 0.0058 | 0.0013 | 0.0172 | 0.0147 |
| | HRI | 0.2072 | 0.0231 | 0.025 | 0.0805 | 1.4426 | 1.3115 | 0.8577 | 0.0098 |
| | | | | **Child** | | | | | |
| Spinach | DIM | 0.0095 | 0.008 | 0.0086 | 0.0129 | 0.0066 | 0.0015 | 0.0197 | 0.0169 |
| | HRI | 0.2382 | 0.0265 | 0.0288 | 0.0925 | 1.6584 | 1.5076 | 0.986 | 0.0113 |

**Table 8.** *Cont.*

| Vegetables | | Cu | Fe | Zn | Mn | Pb | Cd | Ni | Cr |
|---|---|---|---|---|---|---|---|---|---|
| **Sector Z (Alipur)** | | | | | | | | | |
| **Adult** | | | | | | | | | |
| *Adult* | | | | | | | | | |
| Cabbage | DIM | 0.0055 | 0.0082 | 0.0016 | 0.0061 | 1.6584 | 0.0019 | 0.0023 | 0.0144 |
| | HRI | 0.1371 | 0.0275 | 0.0052 | 0.0438 | 1.1148 | 1.941 | 0.1149 | 0.0096 |
| *Child* | | | | | | | | | |
| Cabbage | DIM | 0.0063 | 0.0095 | 0.0018 | 0.0071 | 0.0051 | 0.0022 | 0.0026 | 0.0165 |
| | HRI | 0.1575 | 0.0316 | 0.006 | 0.0504 | 1.2815 | 2.2313 | 0.1321 | 0.011 |
| *Adult* | | | | | | | | | |
| Cauliflower | DIM | 0.0073 | 0.0047 | 0.0043 | 0.0018 | 0.0023 | 0.0005 | 0.0108 | 0.0073 |
| | HRI | 0.183 | 0.0157 | 0.0142 | 0.0127 | 0.5653 | 0.4721 | 0.5414 | 0.0049 |
| *Child* | | | | | | | | | |
| Cauliflower | DIM | 0.0084 | 0.0054 | 0.0049 | 0.0021 | 0.0026 | 0.0005 | 0.0124 | 0.0084 |
| | HRI | 0.2103 | 0.018 | 0.0164 | 0.0146 | 0.6498 | 0.5428 | 0.6224 | 0.0056 |
| *Adult* | | | | | | | | | |
| Radish | DIM | 0.0059 | 0.0128 | 0.0057 | 0.0036 | 0.0052 | 0.0009 | 0.0151 | 0.0023 |
| | HRI | 0.1478 | 0.0428 | 0.0191 | 0.0259 | 1.3010 | 0.918 | 0.7541 | 0.0015 |
| *Child* | | | | | | | | | |
| Radish | DIM | 0.0068 | 0.0147 | 0.0066 | 0.0042 | 0.0060 | 0.0011 | 0.0173 | 0.0026 |
| | HRI | 0.1699 | 0.0491 | 0.022 | 0.0297 | 1.4956 | 1.0554 | 0.8669 | 0.0018 |
| *Adult* | | | | | | | | | |
| Turnip | DIM | 0.0036 | 0.0039 | 0.0041 | 0.0105 | 0.0044 | 0.0005 | 0.0104 | 0.0062 |
| | HRI | 0.0911 | 0.0131 | 0.0137 | 0.0748 | 1.0899 | 0.5194 | 0.5225 | 0.0042 |
| *Child* | | | | | | | | | |
| Turnip | DIM | 0.0042 | 0.0045 | 0.0047 | 0.012 | 0.0050 | 0.0006 | 0.012 | 0.0072 |
| | HRI | 0.1048 | 0.0151 | 0.0157 | 0.0859 | 1.2529 | 0.597 | 0.6006 | 0.0048 |
| *Adult* | | | | | | | | | |
| Tinda | DIM | 0.0047 | 0.003 | 0.0067 | 0.0118 | 0.0054 | 0.0006 | 0.0128 | 0.0071 |
| | HRI | 0.1167 | 0.0101 | 0.0223 | 0.0841 | 1.3574 | 0.6453 | 0.6379 | 0.0047 |
| *Child* | | | | | | | | | |
| Tinda | DIM | 0.0054 | 0.0035 | 0.0077 | 0.0135 | 0.0062 | 0.0007 | 0.0147 | 0.0081 |
| | HRI | 0.1342 | 0.0116 | 0.0256 | 0.0967 | 1.5604 | 0.7418 | 0.7333 | 0.0054 |
| *Adult* | | | | | | | | | |
| Carrot | DIM | 0.0977 | 0.0013 | 0.0028 | 0.0047 | 0.0039 | 0.001 | 0.0112 | 0.0115 |
| | HRI | 0.0045 | 0.0043 | 0.0093 | 0.0333 | 0.9758 | 0.9548 | 0.5613 | 0.0077 |
| *Child* | | | | | | | | | |
| Carrot | DIM | 0.0045 | 0.0015 | 0.0032 | 0.0054 | 0.0045 | 0.0011 | 0.0129 | 0.0132 |
| | HRI | 0.1123 | 0.0049 | 0.0107 | 0.0383 | 1.1217 | 1.0976 | 0.6453 | 0.0088 |
| *Adult* | | | | | | | | | |
| Lettuce | DIM | 0.0052 | 0.0073 | 0.0078 | 0.0086 | 0.0052 | 0.0008 | 0.0146 | 0.0126 |
| | HRI | 0.1288 | 0.0243 | 0.0261 | 0.0611 | 1.3049 | 0.7817 | 0.7279 | 0.0084 |
| *Child* | | | | | | | | | |
| Lettuce | DIM | 0.0059 | 0.0084 | 0.009 | 0.0098 | 0.0060 | 0.0009 | 0.0167 | 0.0144 |
| | HRI | 0.1481 | 0.028 | 0.03 | 0.0702 | 1.5001 | 0.8986 | 0.8367 | 0.0096 |

**Table 8.** *Cont.*

| Vegetables | | Cu | Fe | Zn | Mn | Pb | Cd | Ni | Cr |
|---|---|---|---|---|---|---|---|---|---|
| **Sector Z (Alipur)** | | | | | | | | | |
| **Adult** | | | | | | | | | |
| Adult | | | | | | | | | |
| Colocasia | DIM | 0.006 | 0.0024 | 0.0021 | 0.0059 | 0.0040 | 0.0009 | 0.0157 | 0.0125 |
| | HRI | 0.1489 | 0.008 | 0.0069 | 0.042 | 1.0125 | 0.939 | 0.7848 | 0.0083 |
| Child | | | | | | | | | |
| Colocasia | DIM | 0.0068 | 0.0028 | 0.0024 | 0.0068 | 0.0047 | 0.0011 | 0.018 | 0.0144 |
| | HRI | 0.1711 | 0.0092 | 0.008 | 0.0482 | 1.1639 | 1.0795 | 0.9022 | 0.0096 |

Besides these, Zinc, Pb, Cd, and Mn should be further scrutinized in derived products from the study crops and livestock, particularly meat, milk, and leafy vegetables to determine considerable food safety problems. Besides these, wastewater reuse may cause environmental risks along with health diseases in humans [24]. The results (Table 3) exhibited that observed heavy metals values are above the suggested permissible limit by WHO and described that the human population living in sector X (Kot addu), Sector-Addu), sector Y (Muzaffargarh), and sector Z (Alipur) are using low-quality vegetables. The results for DIM for the ingestion pathway in three study sites are shown in (Tables 6–8) for both children and adults. The results showed that these values were slightly above the reference dose as recommended by WHO or other international bodies. Our results are in agreement with the report of [45]. The HIR indices for the heavy metal consumption by vegetables in the study areas can be ranked in the order cadmium > lead concentration > nickel > chromium for adults, and values of lead were greater for the children living in three study sites and were similar to the finding of [10]. Results of the present investigation demonstrated that Cd, Zn, Fe, Mn, and nickel may cause cancer in humans and could be a major health risk. These findings are in line parallel to [39].

**5. Conclusions**

In the present investigation, the analysis described that soil irrigated with municipal and industrial wastewater had alterations in almost all of the physicochemical parameters. Vegetable crops irrigated with wastewater have considerable amounts of heavy metals, some of which could be harmful to adults' and children's health. These heavy metal transfers from soil to vegetables are significantly driven by soil chemical reactions (pH) and soil cations exchange capacity. Owing to the problem of the bulk of sewage, industrial and domestic wastewater disposal, there is a great problem in Pakistan. It is recommended from the present research work outcomes that wastewater should be pretreated before being used for irrigation of vegetable crops. However, as sources of organic matter, sewage wastewater uses cannot be neglected. This study highlights the primary health issues caused by the use of sewage wastewater for irrigation of vegetable crops and demark the drastic effects of heavy metals in soils and water as well as in vegetables and human populations consuming such poor foodstuff, but also suggests that industrial waste could be used in the best way at commercial level for the country's economic improvement. In Pakistan, the demand for water and protein sources is very high. It has been estimated that more than 12 billion rupees are paid for water resources, and more than 35 million rupees are paid internationally for protein sources (lysine and glutamic acid), which affect the economy of Pakistan. However, the findings of the present research work will be significantly useful in terms of balancing the country's payment along with creating an opportunity for the fermentation industry in Pakistan by using sewage wastewater and some micro-organisms (*Corny bacterium Glutamicum*). By this, we can synthesize amino acids like lysine and glutamic acid at the local scale and can pretreat the sewage waste for irrigation purposes. Therefore, the present study was organized to focus the attention

of scientists, health concerns, and government waste management authorities to develop local processes and guidelines to enhance lysine and glutamic acid production by using industrial effluent and making wastewater suitable for irrigation drives. This research also reveals that the potential human health risks involved in the consumption of vegetables grown with untreated wastewater should not be neglected. As a result, monitoring the wastewater quality used for growing vegetables is crucial to reduce human health risk.

**Author Contributions:** Conceptualization, H.U.; Data curation, M.T.A.; Formal analysis, M.F. and I.A.; Funding acquisition, M.A.W., M.F.K. and A.B.; Investigation, M.T.A. and F.F.; Methodology, M.T.A.; Resources, H.U., F.F. and Z.U.; Software, M.F.; Supervision, H.U.; Validation, I.A.; Writing—original draft, M.T.A.; Writing—review and editing, H.U., F.F., I.A., M.A.W., M.F.K., A.B. and Z.U. All authors have read and agreed to the published version of the manuscript.

**Funding:** This research received no external funding.

**Institutional Review Board Statement:** Not applicable.

**Informed Consent Statement:** Not applicable.

**Data Availability Statement:** All the relevant data are provided in the article.

**Acknowledgments:** The authors express their sincere appreciation to the Research Supporting Project number (RSP2023R466), King Saud University, Riyadh, Saudi Arabia.

**Conflicts of Interest:** The authors declare no conflict of interest.

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
