# Peer review of "Bioaccumulation and Mobility of Heavy Metals in the Soil-Plant System and Health Risk Assessment of Vegetables Irrigated by Wastewater"

_sustainability, doi:10.3390/su152115321_

Round 1

Reviewer 1 Report

It is more than evident that the scarcity of drinking water is a reality. For this reason, this manuscript may be of great relevance for understanding the possibility of using sewage in food production. However, the authors were redundant and did not deepen the discussion of the good results presented. Presenting, in general, a description of them.

Major considerations:

1 - The introduction is redundant, repeating information. I believe that the authors should be more objective, and may have the following sequential theme: (i) the scarcity of potable water and the need for food production; (ii) use of sewage for food production; (iii) risk of this form of food production for the environmental quality (soil), the food produced (vegetables) and the human being (suggestion). Subsequently presenting the main objective of the work. In addition, the originality of this type of analysis is not a justification for carrying out the work. It can be presented in materials and methods, but should not be used as a justification for carrying it out. In addition, I believe it should be noted that this work evaluates some inorganic contaminants present in sewage. But several others may be present and pose a great risk to human health. I know that the work has a focus on heavy metals, but I understand that we should always make this as clear as possible for the reader.;

2 - Material and methods should be better explained. Citing the authors of the methodologies used, always seeking to synthesize this information. Topic 2.4, for example, in addition to also including the determination, is very similar for the three analytical matrices evaluated. Statistical analysis is part of the materials and methods and should be better explained. For example, was the ANOVA performed to compare collection sites or plant species?

3 - The topic results and discussion is very descriptive. Often repeating information that can be verified in the table. In addition, I believe that some static analyzes can be carried out to show the correlation, or not, of the transfer of metal concentration throughout the process.

Minor considerations:

4 - Introduce the meaning of abbreviations before them. Pay attention to the summary;

5 - On page 4, the paragraph "Vegetables are crucial for ... serious human diseases when such plant-based foodstuffs are used [6]." it's verbose. I understand the importance of the quality of the water consumed to maintain the quality of life of human beings. However, the objective of the work is not to treat the water. But using low quality water to produce food;

6 - Item 2.1. The first sentence has to be moved to the end of the introduction;

7 - What equipment was used to determine total dissolved solids?

8 - How was the grinding of the samples performed?

9 - What is the default option of SPSS for ANOVA? Can anyone who does not use this software do the same analysis?

10 - Table 1. Soil variables?

11 - How to talk about enrichment of organic matter in the soil without two collection times?

12 - item 4.3. Were the values for all species averaged?

13 - How was the HMTF (item 4.4.) calculated? what is ANOVA comparing? Difference between sampling sites or between plant species?

14 - Table 5. Why were other variables not treated in the same way? Each vegetable separately. The authors have to explain this in the text and do not describe the results that are presented in the table. You must interpret them and explain the pattern (or lack thereof) found.

Author Response

Query response of Reviewer-i

Query no.

Details Required

Author’s Response

Q1

The introduction is redundant, repeating information. I believe that the authors should be more objective, and may have the following sequential theme: (i) the scarcity of potable water and the need for food production; (ii) use of sewage for food production; (iii) risk of this form of food production for the environmental quality (soil), the food produced (vegetables) and the human being (suggestion). Subsequently presenting the main objective of the work. In addition, the originality of this type of analysis is not a justification for carrying out the work. It can be presented in materials and methods, but should not be used as a justification for carrying it out. In addition, I believe it should be noted that this work evaluates some inorganic contaminants present in sewage. But several others may be present and pose a great risk to human health. I know that the work has a focus on heavy metals, but I understand that we should always make this as clear as possible for the reader.

Introduction part is corrected and focused on main objectives. Highlighted the reasons, for which this study is conducted. Described, why wastewater is preferred by local people for irrigation purposes, impact of heavy metals accumulation in vegetables via wastewater irrigation. Introduction part is reduced and summarized. The repeated lines, sentences or contents have been removed.

The part sentences and repeated words list is attached in anaxture-1.

Line No. 22-128

Q2

Material and methods should be better explained. Citing the authors of the methodologies used, always seeking to synthesize this information. Topic 2.4, for example, in addition to also including the determination, is very similar for the three analytical matrices evaluated. Statistical analysis is part of the materials and methods and should be better explained. For example, was the ANOVA performed to compare collection sites or plant species?

Part: Materials and Methods have been rectified accordingly, as suggested by query. The paragraphs text positions were changed to their relevant place in term of digestion methods procedure.

Line No: Line No. 284-469

Q3

The topic results and discussion is very descriptive. Often repeating information that can be verified in the table. In addition, I believe that some static analyzes can be carried out to show the correlation, or not, of the transfer of metal concentration throughout the process.

Results section has been rectified. ANOVA  (analysis of variance) were performed and results revised accordingly. Table 1,2 and 3 has been changed. As the ANOVA results added.  

Line No.414-439 correlation

Q4

MINOR

Q5

1.       Introduce the meaning of abbreviations before them. Pay attention to the summary;

2.       On page 4, the paragraph "Vegetables are crucial for ... serious human diseases when such plant-based foodstuffs are used [6]." it's verbose. I understand the importance of the quality of the water consumed to maintain the quality of life of human beings. However, the objective of the work is not to treat the water. But using low quality water to produce food;

3.        Item 2.1. The first sentence has to be moved to the end of the introduction;

4.        What equipment was used to determine total dissolved solids?

5.       How was the grinding of the samples performed?

6.        What is the default option of SPSS for ANOVA? Can anyone who does not use this software do the same analysis?

7.       Table 1. Soil variables?

8.        How to talk about enrichment of organic matter in the soil without two collection times?

9.        item 4.3. Were the values for all species averaged?

10.     How was the HMTF (item 4.4.) calculated? what is ANOVA comparing? Difference between sampling sites or between plant species?

11.    Table 5. Why were other variables not treated in the same way? Each vegetable separately. The authors have to explain this in the text and do not describe the results that are presented in the table. You must interpret them and explain the pattern (or lack thereof) found.

1.       Abstract has been modified accordingly. Focused on objectives, scope of work and Described the results in sequence in short summary.  Line no-20 40

2.       Corrected. Excluded lines represented in annexure-i. 

3.       Moved and corrected. (Line No:184-244)

4.       Corrected and described the instrument name and model. Line No. 190.

5.       Described in methodology part.

6.       Corrected the sentence. (Line No:184-244)

7.       Replaced the Teble 1 with new table. In old table some caption mistakes also exist. Soil variables etc.

8.       It was methodology writing errors. Corrected.

9.       The values are means, expressed in tab le 1, 2, 3. Along with standard deviation, and range minimum and maximum.

10.    Rectified.

11.    Rectified accordingly.

Line No. 285-293

Table Number changed.

Reviewer 2 Report

This paper lacks quality of a paper indented to be for publication. This paper should have good sentence writing. Coherence from sentence to sentence and from paragraph to paragraph are needed. I don't think this paper projects the quality needed to be a review paper or a research paper from a sampling study. The authors might have contents, but they lack the impact of the rhetorical pattern in writing. I have concerns in the grammar in the fourth sentence and onwards. I can see several grammar mistakes in the first page, at least five regions. This goes on in the second page. I feel like authors may not have included the correct result from the cited pages. Overall, this paper need to be rewritten to be qualified as a research paper.

This paper lacks quality of a paper indented to be for publication. This paper should have good sentence writing. Coherence from sentence to sentence and from paragraph to paragraph are needed. I don't think this paper projects the quality needed to be a review paper or a research paper from a sampling study. The authors might have contents, but they lack the impact of the rhetorical pattern in writing. I have concerns in the grammar in the fourth sentence and onwards. I can see several grammar mistakes in the first page, at least five regions. This goes on in the second page. I feel like authors may not have included the correct result from the cited pages. Overall, this paper need to be rewritten to be qualified as a research paper.

Author Response

Response to Reviewer 2 Comments on Manuscript

We would like to express our sincere gratitude to the reviewer for taking the time to thoroughly review our manuscript titled “Bioaccumulation and mobility of heavy metals in soil-plant system and health risk assessment of vegetables irrigated by wastewater" We highly value your feedback and have carefully addressed each of the concerns you raised regarding the quality, coherence, writing style, grammar, and content accuracy of our paper. We believe that our revisions have significantly improved the manuscript and aligned it more closely with the standards expected for publication. We have also tried our best to acknowledge all the comments of the other two reviewers. Below, we outline the steps we have taken to address your comments:

  1. Quality and Sentence Writing:

We have meticulously revised the manuscript to enhance the quality of writing throughout. We have focused on sentence structure, clarity, and precision to ensure a more engaging and concise presentation of our research.

  1. Coherence and Flow:

We have restructured both individual sentences and entire paragraphs to enhance the overall coherence and logical flow of the manuscript. Transitions between sentences and paragraphs have been refined to create a smoother reading experience, enabling readers to better follow the line of thought.

  1. Paper Type and Impact:

We understand the importance of projecting the quality needed for a research paper. We have carefully revised the content and framing of our manuscript to ensure that it accurately reflects the research and presents it in a manner befitting a review paper based on a sampling study. Our aim is to provide valuable insights that contribute meaningfully to the field.

  1. Grammar and Language:

Your concerns about grammar have been taken seriously. We have conducted a thorough review of the manuscript and addressed all grammar mistakes in the areas you identified. Additionally, we have performed an in-depth grammar check on the entire manuscript to eliminate any further errors.

  1. Content Accuracy:

We have revisited the cited pages and cross-referenced our results to ensure their accuracy and alignment with the source material. Any discrepancies have been rectified, and the manuscript now accurately reflects the referenced content.

  1. Rewriting and Qualification:

Based on your feedback, we have undertaken a comprehensive rewriting process to elevate the manuscript to the standards expected of a high-quality research paper. The revisions have focused on enhancing clarity, cohesiveness, and the overall impact of our work.

We believe that these revisions have substantially improved the manuscript, addressing each of the concerns you raised. We thank you once again for your valuable input, which has undoubtedly contributed to the refinement of our work. We look forward to your further assessment of the revised manuscript.

Sincerely,

Reviewer 3 Report

  Title: Bioaccumulation and mobility of heavy metals in soil-plant system and health risk assessment of vegetables irrigated by wastewater

 The present study was designed to elucidate heavy metals contamination of vegetables irrigated with domestic wastewater and associated health risk.

There are some drawbacks in paper which should be addressed before publishing.

 Abstract

What is “one fifty-three”??

Move “The daily ingestion of metals (DIM), transfer factor (TF), and health risk index (HRI) were calculated.” Before “The outcomes of …”

Revise “Spinach exhibited higher values of variance in term of Fe and ranged (0.07- 0.11). TF of cabbage was (0.08 to 0.12) for Fe, cauliflower showed range (Fe= 0.05 to 0.06), radish (0.13 to 0.15), tinda (Fe= 0.02 to 0.04), carrot (0.01 to 0.10). Heavy metal levels in vegetables irrigated with wastewater were override than acceptable boundaries”

 Keywords

What is “1.”

Introduction

The introduction is too long and most of the subjects are general topics or a kind of repeated topic. Revise this section.

The statistics and information on the site of the study are too long in this section summarize

Provide similar research on other plants in other part of the world

Material and methods

Replace “physiognomically” with another word

Replace “physicochemical” with another word in material and method and result sections

Revise section 2.4

First paragraph of Section 2.4 should add to the end of 2.3.1 section

Second paragraph of Section 2.4 should add to the end of 2.3.2 section

Third paragraph of Section 2.4 should add to the end of 2.3.3 section

Section 2.4 “At two separate temperatures, the second digestion was carried out for 180 minutes at 120°C after the first digestion took place for 30 minutes at 50°C” is unclear for me, revise

Section 2.4 “We used to detect the amounts of heavy metals such as Cr, Cd, Cu, Fe, Ni, Zn and Mn” is unclear for me, revise

Result

The analysis of variance is stated in statistical analysis but not provided in result section

Re-write the result section after doing variance analysis

The SD in result text is inserted for some values and not for some. Please insert for all or delete for all.

Author Response

Query response of Reviewer-iii

Query no.

Details Required

Author’s Response

Q1

Abstract ……. What is “one fifty-three”??

Corrected in Abstract----line no.4

Q2

Move “The daily ingestion of metals (DIM), transfer factor (TF), and health risk index (HRI) were calculated.” Before “The outcomes of …”

Corrected accordingly. Added the out comes in abstract before The daily ingestion of metals (DIM), transfer factor (TF), and health risk index (HRI)

Line No.27-38

Q3

Revise “Spinach exhibited higher values of variance in term of Fe and ranged (0.07- 0.11). TF of cabbage was (0.08 to 0.12) for Fe, cauliflower showed range (Fe= 0.05 to 0.06), radish (0.13 to 0.15), tinda (Fe= 0.02 to 0.04), carrot (0.01 to 0.10). Heavy metal levels in vegetables irrigated with wastewater were override than acceptable boundaries”

Corrected accordingly.

Line No.27-38

Q4

Keywords:…..What is “1.”

Formatting errors, corrected 1’ assigned to heading introduction.

Introduction

Q5

The introduction is too long and most of the subjects are general topics or a kind of repeated topic. Revise this section.

Introduction part is corrected and focused on main objectives. Highlighted the reasons, for which this study is conducted. Described, why wastewater is preferred by local people for irrigation purposes, impact of heavy metals accumulation in vegetables via wastewater irrigation. Introduction part is reduced and summarized. The repeated lines, sentences or contents have been removed.

Line No. 22-128

Q6

The statistics and information on the site of the study are too long in this section summarize

Reduced accordingly as suggested by reviewer

Line No. 22-128

Q7

Provide similar research on other plants in other part of the world

Provided the information and review of similar research on other plants in other parts of word.

Material and methods

Q9

Replace “physiognomically” with another word

Replaced this word by ‘ apparently’

Line no. 147

Q10

Replace “physicochemical” with another word in material and method and result sections

Corrected as suggested by Reviewer.

Line No.206

Q11

Revise section 2.4

Corrected accordingly. Merged the digestion procedure in relevant part of wastewater, soil and vegetation.

Q12

First paragraph of Section 2.4 should add to the end of 2.3.1 section

Corrected accordingly.

Line No.217 to line No. 225

Q13

Second paragraph of Section 2.4 should add to the end of 2.3.2 section

Corrected accordingly. Line No.226

Q14

Third paragraph of Section 2.4 should add to the end of 2.3.3 section

Corrected accordingly.

Line No.235-239

Q15

Section 2.4 “At two separate temperatures, the second digestion was carried out for 180 minutes at 120°C after the first digestion took place for 30 minutes at 50°C” is unclear for me, revise

Corrected accordingly.

Line No. 217 to 224

Q16

Section 2.4 “We used to detect the amounts of heavy metals such as Cr, Cd, Cu, Fe, Ni, Zn and Mn” is unclear for me, revise

Corrected accordingly.

Line No.202-204

Result

Q17

The analysis of variance is stated in statistical analysis but not provided in result section

ANOVA, results provided in results section in tables and described in text for elaborating the findings.

Line no.279-283

Q18

Re-write the result section after doing variance analysis

Corrected accordingly. Table 1,2 and 3 were newly generated along with ANOVA result values.

Line No.284-244

Q19

The SD in result text is inserted for some values and not for some. Please insert for all or delete for all

Corrected accordingly.

Table 1, 2, 3. SD values are given

Round 2

Reviewer 1 Report

The authors responded to the interventions made. With the exception of the inclusion of statistical analyses in the material and methods, which he still believes should be done.

Author Response

Query response of Reviewer-i

Query no.

Details Required

Author’s Response

Q1

The authors responded to the interventions made. With the exception of the inclusion of statistical analyses in the material and methods, which he still believes should be done.

 The result part was modified after first revision. Tables were rewritten and ANOVA results were incorporated (f-values and P-values along with level of significance). In the main file, for justification whole table contents of ANOVA were expressed in the revised manuscript.  

Reviewer 2 Report

I do have concerns still on the english writings. Line number 33 and 34 say .".. parallel values of HRI...". I do not understand what it is compared to. Line number 68 and 69 still have problems with comas, capital letters etc. Please refer line number 114 and see how Durum wheat and Soft wheat are written. I do have concerns on citations. For example, citation [1] is not appropriate for the sentence you mentioned. Also, look at the citation [13]. It is not appropriately cited for the context. I am sure, lot of citations should have been mentioned inappropriately. I do not suggest this for publication with these concerns. I do not think the paper has improved in quality for publication in Sustainability. 

English writing is still a concern.

Author Response

Query response of Reviewer-ii

Query no.

Details Required

Author’s Response

Q1

I do have concerns still on the english writings. Line number 33 and 34 say .".. parallel values of HRI...". I do not understand what it is compared to. Line number 68 and 69 still have problems with comas, capital letters etc. Please refer line number 114 and see how Durum wheat and Soft wheat are written. I do have concerns on citations. For example, citation [1] is not appropriate for the sentence you mentioned. Also, look at the citation [13]. It is not appropriately cited for the context. I am sure, lot of citations should have been mentioned inappropriately. I do not suggest this for publication with these concerns. I do not think the paper has improved in quality for publication in Sustainability. 

Dear Sir/Madam, Thank you for your valuable comments. We have now tried our best to improve the grammar of our manuscript, and also run different grammar software. Sir, your appreciated suggestions really help us to improve the quality of our work. The whole of the quires was adjusted and expressed in the revised template. Hope you will consider our efforts this time and shower your blessing on us. Waiting eagerly for your kind response.

References were cross-checked and expressed for justification.

Reviewer 3 Report

Thanks

The authors have corrected all the proposed suggestions

Author Response

Respected Reviewer,

Thank you so much for the approval of our manuscript for publication in Sustainability. God Bless You .

Kind Regards

Round 3

Reviewer 2 Report

I think so far the authors took care of the comments